# A Comparative Analysis of NT-proBNP Levels in Pregnant Women and the Impact of SARS-CoV-2 Infection: Influence on Birth Outcome

**DOI:** 10.3390/diseases12010010

**Published:** 2023-12-30

**Authors:** Carmen-Ioana Marta, Marius Craina, Razvan Nitu, Anca Laura Maghiari, Simona-Alina Abu-Awwad, Lioara Boscu, Mircea Diaconu, Catalin Dumitru, George Dahma, Ionela-Iasmina Yasar, Katalin Babes

**Affiliations:** 1Doctoral School, Faculty of Medicine and Pharmacy of Oradea, University of Oradea, 410087 Oradea, Romania; silaghi.carmen@gmail.com; 2Clinic of Obstetrics and Gynecology, “Pius Brinzeu” County Clinical Emergency Hospital, 300723 Timisoara, Romania; mariuscraina@hotmail.com (M.C.); diaconu.mircea@umft.ro (M.D.); dumitru.catalin@umft.ro (C.D.); george.dahma@umft.ro (G.D.); 3Department of Obstetrics and Gynecology, Faculty of Medicine, “Victor Babes” University of Medicine and Pharmacy, 300041 Timisoara, Romania; 4Department I—Discipline of Anatomy and Embryology, “Victor Babes” University of Medicine and Pharmacy, 300041 Timisoara, Romania; boscu.anca@umft.ro; 5Doctoral School, “Victor Babes” University of Medicine and Pharmacy, 300041 Timisoara, Romania; alina.abuawwad@umft.ro (S.-A.A.-A.); lioara.boscu@umft.ro (L.B.); dryasariasmina@gmail.com (I.-I.Y.); 6Department IX: Surgery I, Faculty of Medicine, “Victor Babes” University of Medicine and Pharmacy, 300041 Timisoara, Romania; 7Faculty of Medicine and Pharmacy of Oradea, University of Oradea, 410087 Oradea, Romania; piszekati@yahoo.co.uk; 8Clinical County Emergency Hospital of Oradea, 410167 Oradea, Romania

**Keywords:** NT-proBNP, COVID-19, pregnancy complications, cardiovascular diseases

## Abstract

Background: The cardiac biomarker NT-proBNP is released by the ventricles in response to increased cardiac wall tension, showing cardiac activity in heart failure. The primary objective of this comparative study was to analyze the variations of NT-proBNP levels among pregnant patients and to determine the potential influence of SARS-CoV-2 infection on these values. Secondly, the study focused on NT-proBNP levels and their influence on the type of birth. Methods: Blood samples were taken from 160 pregnant mothers in order to determine, through the solid-phase enzyme-linked immunosorbent assay (ELISA) method, the NT-proBNP concentrations from the plasma. The cohort was separated into two distinct groups based on SARS-CoV-2 diagnostic results: negative to the infection, and positive to the infection. Results: The SARS-CoV-2-positive group of patients presented with higher levels of NT-proBNP and had higher rates of cesarean sections. (4) Conclusions: Our research highlights the crucial relationship between elevated NT-proBNP values and the mode of giving birth, natural delivery or cesarean section, and also the influence of SARS-CoV-2 viral infection and this biomarker.

## 1. Introduction

The consequences and outcomes of COVID-19 on pregnant patients are increasing, and there are still many question marks about the viral infection associated with pregnancy [1]. COVID-19 infection is strongly linked to significant maternal mortality and morbidity [2]. According to recent research, pregnant patients with SARS-CoV-2 viral infection had a greater risk of ICU hospitalization, mechanical ventilation, and mortality when compared to pregnant women who did not have this kind of viral infection, as well as non-pregnant women [3,4,5].

Puerperal periods and pregnancy constitute a specific clinical context because of their unique hormonal and immunological modifications [6]. Another viral disease such as infection with the human immunodeficiency virus (HIV) while pregnant has been shown to increase the rate of post-operative complications after a cesarean section in accordance with the severity of the infection. Even though there are a lot of medications considered to be teratogenic, especially if they are administered in the first trimester of pregnancy, antiretroviral treatment for HIV in the first trimester has been shown not to increase the risk of congenital abnormalities [7,8]. Pregnancy necessitates a delicate balance between natural immunological tolerance for the fetus and sustaining an immune response against microbial infections. The specific type of this is determined by the gestational age [9]. All of this is essential to accomplish correct fetal development at specific gestational weeks. The alterations are not only visible in the endometrium but also throughout the body, which may impact the course of the viral infection if detected during a certain gestational week. The immunological state in pregnant women includes increased activity of plasmacytoid dendritic cells, natural killer (NK) cells, and monocytes, resulting in an increased response to viral infection [9,10,11].

The cardiac biomarker NT-proBNP is released by the ventricles in response to increased cardiac wall tension, showing cardiac activity in heart failure [12]. Its measured levels are highly correlated with the severity of heart failure, making it a valuable biomarker for heart failure diagnosis, treatment, and prognosis [13,14]. Nevertheless, persistently higher NT-proBNP values in response to sustained cardiac stress can have negative consequences on the blood vessels and heart, such as vascular remodeling, myocardial fibrosis, and cardiac hypertrophy [15,16]. The most significant diagnostic and prognostic marker in heart failure is the value of NT-proBNP. This marker has also been shown to be of clinical importance in other cardiac diseases and infections [17,18,19]. The relationship between natriuretic peptides and SARS-CoV-2 is still unclear. This is the reason why many theories have appeared in order to shed light on this subject. Higher levels of NT-proBNP in COVID-19 viral infection can be attributed to myocardial damage, inflammation, angiotensin-converting enzyme (ACE2) deterioration, and cardiac function degradation, associated with acute heart failure [20].

The primary objective of this comparative study was to analyze the variations in NT-proBNP levels among pregnant patients and to determine the potential influence of SARS-CoV-2 infection on these values. Secondly, this study focused on NT-proBNP levels and their influence on the type of birth. NT-proBNP, a marker commonly associated with cardiac stress, has been the subject of several studies in relation to various health conditions. Given the general impact of pregnancy on cardiovascular health, combined with the potential complications caused by SARS-CoV-2 infection, understanding the relationship between these variables is of utmost importance. This research seeks to highlight whether pregnant individuals with SARS-CoV-2 infection demonstrate significantly different NT-proBNP levels compared to their non-infected counterparts, thereby offering insights into potential cardiac implications and guiding better clinical decision-making when it comes to managing COVID-19 viral infection in pregnant patients.

## 2. Materials and Methods

### 2.1. Study Population/Sample Selection

Blood samples were taken from 160 pregnant mothers in order to determine NT-proBNP concentrations. The cohort was separated into two distinct groups based on SARS-CoV-2 diagnostic results: negative for the infection and positive for the infection. Group 1 includes 83 pregnant participants with no detectable SARS-CoV-2 infection, whereas Group 2 includes 77 pregnant participants with confirmed SARS-CoV-2 viral infection. This observational study was meticulously conducted starting from 1 January 2021 to 31 December 2022. It is important to mention that all of the studied patients were not vaccinated against SARS-CoV-2 viral infection because of their skepticism toward possible adverse reactions during pregnancy or in a future possible pregnancy.

### 2.2. RT-PCR SARS-CoV-2 Analysis

SARS-CoV-2 viral infection identification was obtained through sampling of the nasopharynx and oropharynx with two separate swabs, by a medical professional that had adequate equipment in order to avoid tampering with the collected samples or risking contracting COVID-19 infection. The collected samples were packed in a sterile recipient with liquid medium and transported to the laboratory.

COVID-19 viral infection was identified through reverse transcription polymerase chain reaction (RT-PCR). In the identification process through the RT-PCR method, firstly, with a reverse transcription enzyme, the viral RNA is transformed into complementary DNA (cDNA). Subsequently, specific parts of this DNA are exponentially amplified using a thermo-stable polymerase. The amplification process is monitored in real time, and due to the use of specific probes and primers, it allows for the precise detection and quantification of the viral target. The presence of the virus is confirmed through the identification of specific fluorescent signals emitted during the amplification process.

### 2.3. NT-proBNP Analysis

In order to measure the levels of NT-proBNP, venous blood samples were collected from each individual in the studied groups by inserting a sterile needle into the vein and extracting the blood into a collecting tube or a syringe. After an adequate probe was obtained, the sample was transported to the laboratory and then centrifuged. The solid-phase enzyme-linked immunosorbent assay (ELISA) method was used for the evaluation of NT-proBNP levels from the plasma.

In order to assess the long-term effects, for patients in Group 2, blood samples were systematically collected at two distinct postpartum intervals for measuring NT-proBNP levels. The first tests were at 6 months postpartum, and subsequent testing was performed at the 12-month postpartum mark. These timepoints were chosen to provide a comprehensive understanding of the temporal changes in NT-proBNP levels within the first year after delivery, ensuring an accurate representation of the patients’ cardiovascular health trajectory during this critical period.

### 2.4. Inclusion and Exclusion Criteria

Participants were meticulously selected and corresponded to the well-defined inclusion and exclusion criteria established for the study.

Inclusion criteria for participant selection: pregnant patients who gave birth in our hospital (by vaginal or cesarian section delivery); patients between the ages of 18 and 45; patients who were willing to engage willingly and supply blood samples for analysis; patients who had tested positive or negative for SARS-CoV-2 in the two weeks prior to the study, as validated by PCR; women with no history of cardiovascular disease or other illnesses that might alter NT-proBNP levels; pregnant women who completed informed permission forms for our study; patients who received prenatal care and had frequent prenatal checkups.

Exclusion criteria for participant selection: women who were not pregnant or who were in the post-partum period; patients who experienced substantial pregnancy problems, such as severe preeclampsia or uncontrolled gestational diabetes; patients who had taken drugs in the two weeks prior to the study that may have influenced NT-proBNP levels; pregnant patients who had a history of drug abuse; women who had a history of psychiatric disorders; patients who participated in other clinical studies during the 6 months prior to the study; women who refused to participate in the study and did not sign an informed consent form; patients who previously experienced unfavorable reactions to blood draws; pregnant patients who did not perform regular examinations during the pregnancy.

### 2.5. Ethical Considerations

For this study, rigorous ethical standards were used. All participants submitted written consent prior to data collection after being adequately informed about the study’s aims, methodology, potential risks, and benefits. Personal identifiable patient information was deleted or appropriately encrypted to protect the anonymity and confidentiality of the participants’ information. Prior to beginning, the institutional review board regranted ethical permission (approval No. 6/15 January 2021).

Participants in this study were given information regarding the study’s purpose, which is to understand any alterations in NT-proBNP levels throughout pregnancy and the possible modifications to these levels induced by SARS-CoV-2 infection. Participants were informed of the procedures involved, namely blood drawing, and were aware of minor hazards, such as brief soreness at the puncture site. Their autonomy, confidentiality, and ability to withdraw at any stage throughout the trial without jeopardizing their routine treatment were all honored.

### 2.6. Statistical Analysis

The data obtained in the study were processed with GraphPad Prism software, version 5. Descriptive statistics was used for summarizing the data, while *t*-tests were used for group comparisons. The statistical tests performed were two-tailed. Furthermore, the data were presented as mean values with standard deviations (expressed as mean ± standard deviation, or SD). The z-test was used for binomial proportions in order to compare the observed percentages between the two studied groups to determine if there was a statistically significant difference in the studied proportions of the two groups. A small *p*-value (less than 0.05) suggests a greater difference between the proportions, clarifying if the observed variance is likely due to random chance or represents a true difference between the cohorts.

## 3. Results

### 3.1. Demographic Distribution of Patients

A comprehensive demographic analysis of patients segregated into two distinct groups, Group 1 and Group 2 (Table 1), was conducted. The age distribution reveals three categories: those below 25 years old, between 25 and 34 years old, and above 35 years old. The educational background of the patients is divided into four classifications: those without formal education, patients with primary education, high school graduates, and those with higher education. No significant difference can be observed in the age distribution and educational background classifications. The occupation profile comprises three major classifications: unemployed individuals, students, and employed individuals. A remarkable difference is observed in the employed category, where Group 2 surpasses Group 1. Additionally, the classification of living either in an urban or rural environment is also presented, indicating a balanced distribution between the two groups. As for the values of NT-proBNP, statistical significance can be found in all of our studied demographic categories. The *p*-value for each demographic parameter indicates the statistical significance of the observed differences, with a value less than 0.05 indicating significant differences.

### 3.2. Distribution of Patients According to the Type of Birth

Additionally, we conducted a comparative analysis of the type of birth distribution between the two groups, distinguishing between cesarean section and natural birth (Table 2) and the NT-proBNP values. An important difference is evident in the proportions between the two groups. Group 2 has a notably higher percentage of patients who gave birth through cesarean section compared to Group 1. As opposed to this, Group 1 shows a predominant inclination toward natural birth, in contrast with the figures for Group 2. The associated *p*-values, being less than 0.0003 for both categories, underscore the statistically significant disparities between the two groups concerning their mode of giving birth and their NT-proBNP levels. In Group 1, the predominant inclination toward natural birth can be attributed to promoting natural delivery as the favored type of birth. Medical practitioners and healthcare facilities emphasize the benefits and physiological importance of natural birth, encouraging a more organic approach to childbirth. Furthermore, those who underwent a cesarean section within this group had well-grounded medical indications, suggesting that this surgical intervention was used only for cases where it was medically necessary.

The greater occurrence of cesarean sections in Group 2 is closely connected to the difficulties associated with COVID-19 viral infection. Many healthcare practitioners had to change their approach due to the virus’s possible hazards and health repercussions. In cases when the mother’s health was put at risk by the virus, an emergency cesarean section was performed in order to prevent any danger that might have influenced the pregnant patient or the fetus’ well-being.

### 3.3. Comparison of NT-proBNP Levels in Patients

Table 3 provides important statistical descriptors for each cohort as presented below:

### 3.4. Evolution of NT-proBNP Levels in COVID-19-Positive Patients

Table 4 provides an analytical image of the trajectory of NT-proBNP levels in Group 2 after the postpartum period. At 6 months postpartum, the majority of patients had decreased NT-proBNP levels, while a minority showed elevated values. Intriguingly, of the cohort that registered elevated levels at the 6-month interval, the 12-month assessment unveiled that a substantial number transitioned to reduced NT-proBNP levels. Nonetheless, a small number of patients persisted with elevated values beyond the 12-month evaluation, denoting potential long-term cardiac concerns and implications. The statistical assessment, reflected by a *p*-value below 0.001, accentuates the marked significance in NT-proBNP level modifications between the immediate post-birth period and the successive postpartum evaluations.

### 3.5. Distribution of Associated Pathologies

Table 5 provides a comparative overview of various associated pathologies observed in the two distinct studied groups. It reveals an intriguing pattern of medical conditions ranging from hypothyroidism to autoimmune diseases. While both groups show occurrences of these pathologies, the distribution varies subtly between them. For example, conditions such as hypothyroidism and obstructive sleep apnea are present in both groups, but with slightly higher percentages in Group 1. Thrombophilia, on the other hand, is more evenly distributed between the two groups. Interestingly, bronchial asthma appears exclusively in Group 1, hinting at a potential area for further investigation. In contrast, chronic viral infections are more prevalent in Group 2. The presence of autoimmune diseases in both groups, although more common in Group 1, adds another layer of complexity to the overall health profile of the patients.

## 4. Discussion

The wide prospect of medical research and clinical investigations consistently associates pregnant patients with a heightened sensitivity to numerous illnesses. This sensitivity might be assigned to a variety of physiological and immunological changes that occur during pregnancy. Given this pre-existing understanding, especially in light of the current global health crisis highlighted by SARS-CoV-2 infection, it is crucial to emphasize the possible danger that pregnant patients may face regardless of demographic categories such as those presented in the study. Our results reveal that age, educational background, and living environment are not relevant or of importance as a predisposing factor. Statistical significance can be found only by measuring the values of NT-proBNP in the demographic categories. Furthermore, newborns should also be acknowledged as a vulnerable group in contact with SARS-CoV-2 viral infection because of the possibility of early exposure and common physiological links. Therefore, it is imperative for healthcare providers to take into consideration these patients when devising protective strategies and guidelines [21]. In the present study, special attention has been given to pregnant women who tested positive for COVID-19 and their newborns, as we attempted to develop new care strategies for these patients.

The estimated mortality rates for pregnant women with SARS-CoV-2 infection have been demonstrated to vary between 1.35% and 12.3% [22,23,24]. In the present study, there were four deaths among the positive patients (5.19%) of the total patients included in this group.

Clinical symptoms in COVID-19-positive individuals during pregnancy are similar to those seen in non-pregnant people. The most prevalent clinical signs are cough (41% of patients), fever (40% of patients), and dyspnea (21% of patients). It is important to note that pregnant women are more likely to have an asymptomatic COVID-19 infection [4,25].

When it comes to the individual risk factors that have an impact on the mortality rates of pregnant women with this viral infection, a recent meta-analysis assessed the impact of maternal risk factors such as diabetes, obesity, and asthma and came to the conclusion that maternal obesity elevated the risk of death by 2.48% [26]. Of the four deaths in our study, two of the pregnant women had a BMI over 30 kg/m^2^, which reinforces the findings of the meta-analysis presented above.

Another meta-analysis evaluated the impact size of related comorbidities on maternal death in COVID-19-infected patients and discovered an effect size of 0.47 for overweight patients, 0.29 for diabetic mellitus, 0.41 for asthmatic patients, and 0.61 for advanced maternal age [27]. In our study, we encountered a variety of comorbidities, including patients with multiple types of associated diseases. Intriguingly, most of the patients who had associated pathologies in the presented study were SARS-CoV-2-negative patients. Consequently, it becomes challenging to accurately quantify the extent to which diabetes mellitus, asthma, or advanced maternal age influence mortality in patients with positive SARS-CoV-2 infection.

Some studies have reported connections of COVID-19 infection with some cardiovascular complications, for instance, arrhythmias, heart failure, and myocarditis, and can be correlated to our research when talking about higher values of NT-proBNP, which indicates these cardiovascular conditions in the studied group of pregnant patients infected with COVID-19. Individuals with pre-existing cardiovascular pathologies such as heart failure, coronary artery disease, and hypertension, on the other hand, have a higher risk of a severe form of the viral infection as well as a higher mortality rate [28,29].

The placenta’s synthesis of estrogen and progesterone causes alterations in the circulatory system [30]. Circulating blood volume increases and reaches a peak between 32 and 34 weeks of pregnancy [31].

BNP and NT-proBNP levels have been shown to be greater in COVID-19-infected individuals with cardiac damage [32,33]. Some studies have connected cardiac biomarkers, such as NT-proBNP, to an increased risk of death in SARS-CoV-2 patients. Chen et al. imply that NT-proBNP was significantly elevated in deceased COVID-19 patients in comparison with the patients who survived [34]. Deceased women in our study were found to mostly have the highest NT-proBNP values among the studied patients, except for a few cases where the values of NT-proBNP were elevated but not at the peak of our statistical analysis. Therefore, we can argue that deceased SARS-CoV-2-positive patients generally have higher NT-proBNP values than those who survived, but we cannot claim that the highest NT-proBNP values are found in deceased patients.

Caro-Codón et al. suggested that NT-proBNP levels were greater in COVID-19 patients compared to healthy control patients and that higher levels were related to a more severe form of the illness, a higher risk of mortality, and a longer hospitalization. This study observed that NT-proBNP levels decreased in individuals who survived the disease, but remained elevated or had risen in those who did not survive. This study demonstrates that NT-proBNP might be a useful biomarker for predicting sickness severity and outcomes in SARS-CoV-2-infected people [35]. The results of our study support this suggestion.

Furthermore, a study highlighted that patients with higher values of NT-proBNP on the first day of hospitalization had a higher chance of developing complications such as risk of loss of life, longer admission period, and requirement for mechanical ventilation [36]. In addition to this, another study found that patients with pre-existing cardiac disease who had greater levels of NT-proBNP had a higher risk of death during admission to hospital [37].

While our main focus of the study was to highlight if viral infection with COVID-19 increased NT-proBNP values, which results in a higher risk of heart failure in general, greater values of NT-proBNP also had an influence on the birth outcome and increased the rates of cesarean section delivery.

Our results underline the importance of evaluating the patients’ cardiovascular system status, especially in pregnant patients, by monitoring NT-proBNP values, particularly if SARS-CoV-2 viral infection is associated because of its influence on the cardiovascular system and birth outcome.

Strengths and limitations:

The study had several strengths, such as the rigorous selection of the cases included, a well-established study design, and specific attention to a high-risk group of pregnant women with COVID-19 viral infection. In addition to this, because NT-proBNP in pregnant patients in general and in those with SARS-CoV-2 viral infection has been insufficiently studied and may be a critical topic, our results give important information when it comes to adequate prenatal care and associated risk factors for these vulnerable individuals.

While this research has notable strengths, it is imperative to acknowledge certain inherent limitations that might affect its generalizability. Foremost, the temporal constraints, characterized by the relatively brief duration over which the study was conducted, may potentially limit its comprehensive depth. Moreover, its single-center design precludes the comprehensive representation of diverse demographics of pregnant patients, potentially attenuating the study’s breadth. Another significant concern is the sample size; the small number of patients evaluated may not give the validity that a larger, more varied cohort would. These characteristics emphasize the significance of caution when interpreting the data and give clarification in future studies on the relationship between NT-proBNP levels and COVID-19 in the pregnant population.

Our aspiration is to persistently advance with this study within our esteemed institution while simultaneously encouraging shared patient trials with other institutions and distinguished researchers in this specialized domain. We are convinced that broadening the scope of our research is critical in enhancing the legitimacy of our preliminary findings. To provide a more thorough knowledge of the consequences of NT-proBNP levels and their importance in assessing cardiovascular risk among pregnant women with COVID-19 viral infection, we must expand the size of our study group. Furthermore, performing parallel analyses across several research centers will not only increase the validity of our findings but will also ensure their application in a variety of clinical contexts.

## 5. Conclusions

In conclusion, in our study, we meticulously examined the discrepancies between the levels of NT-proBNP in two distinct cohorts of pregnant patients: those who have been diagnosed with COVID-19 infection and those who are not infected.

Furthermore, an important finding of our research highlights the crucial relationship between elevated NT-proBNP and the mode of giving birth, favoring cesarean section.

The findings from our rigorous research are of great importance when it comes to prenatal care, as they offer insights aimed at clarifying and strengthening cardiovascular health strategies for both the expectant mother and the fetus. However, it is critical to recognize the intricacies underlying the effects of NT-proBNP levels on the cardiovascular system.

Ultimately, additional research is needed in order to unravel the complex mechanisms and implications of SARS-CoV-2 infection on the cardiovascular system and its long-term consequences on patients.

## Figures and Tables

**Table 1 diseases-12-00010-t001:** Demographic distribution of patients from Group 1 and Group 2.

	Group 1 (n = 83)	Group 2 (n = 77)	*p* Value
Age			
Under 25 years old	23 patients (27.71%)	23 patients (29.87%)	0.763
NT-proBNP level (pg/dL) Mean ± SD	102.9 ± 104.4	425.1 ± 268.1	<0.0001
Between 25 and 34 years old	46 patients (55.42%)	42 patients (54.54%)	0.912
NT-proBNP level (pg/dL) Mean ± SD	117.3 ± 124.3	498.4 ± 536.3	<0.0001
Over 35 years old	14 patients (16.86%)	12 patients (15.58%)	0.822
NT-proBNP level (pg/dL) Mean ± SD	79.36 ± 8.889	407.8 ± 225.8	<0.0001
Level of education			
No education	13 patients (15.66%)	11 patients (14.28%)	0.804
NT-proBNP level (pg/dL) Mean ± SD	83.38 ± 9.683	456.5 ± 171.5	<0.0001
Primary education	21 patients (25.30%)	25 patients (32.46%)	0.300
NT-proBNP level (pg/dL) Mean ± SD	224.5 ± 323.3	533.1 ± 203.8	0.0003
High school	32 patients (38.55%)	20 patients (25.97%)	0.09
NT-proBNP level (pg/dL) Mean ± SD	218.6 ± 294.4	482.4 ± 154.9	0.0006
Higher education	17 patients (20.48%)	21 patients (27.27%)	0.296
NT-proBNP level (pg/dL) Mean ± SD	162.8 ± 200.6	489.8 ± 154.7	<0.0001
Occupation			
No occupation	23 patients (27.71%)	14 patients (18.18%)	0.154
NT-proBNP level (pg/dL) Mean ± SD	142.3 ± 174.7	596.3 ± 237.3	<0.0001
Student	21 patients (25.30%)	14 patients (18.18%)	0.276
NT-proBNP level (pg/dL) Mean ± SD	147.7 ± 182.3	453.1 ± 158.9	<0.0001
Employed	39 patients (46.98%)	49 patients (63.63%)	0.0166
NT-proBNP level (pg/dL) Mean ± SD	133.6 ± 154.4	502.5 ± 149.8	<0.0001
Living environment			
Urban	43 patients (51.80%)	29 patients (37.66%)	0.0434
NT-proBNP level (pg/dL) Mean ± SD	141.5 ± 166.9	580.3 ± 196.2	<0.0001
Rural	40 patients (48.19%)	48 patients (62.33%)	0.0426
NT-proBNP level (pg/dL) Mean ± SD	143.6 ± 172.9	583.7 ± 482.7	<0.0001

**Table 2 diseases-12-00010-t002:** Distribution of patients from Group 1 and Group 2 according to the type of birth.

Birth Method	Group 1 (n = 83)	Group 2 (n = 77)	*p* Value
Cesarean section	25 patients (30.12%)	57 patients (74.02%)	<0.001
NT-proBNP level (pg/dL) Mean ± SD	174.8 ± 213.1	555.2 ± 461.7	0.0002
Natural birth	58 patients (69.87%)	20 patients (25.97%)	<0.001
NT-proBNP level (pg/dL) Mean ± SD	149.8 ± 180.0	574.9 ± 204.1	<0.001

**Table 3 diseases-12-00010-t003:** Comparison of NT-proBNP levels in COVID-19-negative (Group 1) and -positive (Group 2) pregnant patients.

	Group 1 (n = 83)	Group 2 (n = 77)
Minimum	68.00	49.00
25% Percentile	76.00	275.5
Median	87.00	459.0
75% Percentile	139.0	597.0
Mean	1721	3238
Std. Deviation	308.6	418.2
*p*-value	<0.0001	

**Table 4 diseases-12-00010-t004:** Evolution of NT-proBNP levels in Group 2. * Three of the patients maintained elevated levels of NT-proBNP for more than 12 months postpartum. Note: A *p*-value < 0.001 indicates a statistically significant difference when comparing birth results of those at 6 months and 12 months postpartum.

Timepoint	Total Patients	Reduced Levels	Elevated Levels
6 months postpartum	77 (100%)	58 (75.32%)	19 (24.67%)
12 months postpartum (from elevated group at 6 months)	19 (24.67%)	16 (20.77%)	3 * (3.89%)

**Table 5 diseases-12-00010-t005:** Comparison of associated pathologies in COVID-19-negative (Group 1) and -positive (Group 2) pregnant patients.

Associated Pathologies	Group 1 (n = 83)	Group 2 (n = 77)	*p*-Value
Hypothyroidism	4 (4.81%)	2 (2.59%)	0.4602
Obstructive sleep apnea	2 (2.40%)	1 (1.29%)	0.605
Thrombophilia	12 (14.45%)	9 (11.68%)	0.572
Bronchial asthma	1 (1.20%)	0 (0%)	<0.001
Chronic viral infections	3 (3.61%)	5 (6.49%)	0.40
Autoimmune diseases	3 (3.61%)	1 (1.29%)	0.35

## Data Availability

The data presented in this study are available on request from the first author. The data are not publicly available due to ethical considerations.

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
