# Peer review of "A Comparative Analysis of NT-proBNP Levels in Pregnant Women and the Impact of SARS-CoV-2 Infection: Influence on Birth Outcome"

_diseases, 2023, doi:10.3390/diseases12010010_

Round 1

Reviewer 1 Report

Comments and Suggestions for Authors

1. The method of measuring NT-proBNP concentrations should mention in the method section of abstract.

2. The first two sentences of introduction are unnecessary to mention here.

3. The inclusion and exclusion criteria should present concisely as paragraph (not bullet point). Also, the authors can add subsection for methodology like starting with 1. study design and participants, 2. Inclusion and exclusion criteria, 3. Sample collection and analysis, 4. Statistical analysis, etc, which ever they think appropriate. This structured method will help the readers to follow this study.

4. Similarly, the results can be presented as subsection, for example, 1. Description of the study participants, 2......so on.

5. Please discuss your findings with respect to earlier studies. Give possible reason (s) for your findings in contrast or similarity to others by explaining explicitly what you want to pass across to the scientific audience.

Comments on the Quality of English Language

None.

Reviewer 2 Report

Comments and Suggestions for Authors

Dear Authors,

Personally, I missed the link by using the selected tables as "Level of education, Occupation, Living environment, associated pathologies", most of these are mentioned only once in the results. Furthermore, these are totally not discussed in the discussion. If these selected results are not valid for this comparative analysis, why include all these tables in the manuscript? Also, the part of the underlying or associated diseases seems for me an important point to address. The key conclusion in the role of NT-proBNP levels in pregnant women and COVID-19 is clearly written, although addressing deeper the biological role or mechanism behind this relation ship would be appreciated as a reader.

Comments on the Quality of English Language

Is fine and no editing required.

Reviewer 3 Report

Comments and Suggestions for Authors

Your study even if  deals with  an interesting topic with the aim to analyze the variations of NT-proBNP levels among pregnant patients with and without SARS-CoV-2 infection and to evaluate a  possible influence on the way of birth , it has many limitations and is therefore not suitable for publication. The sample analyzed is numerically small; there is no an appropriate correlation with other confusing maternal and obstetrical risk factors such as diabetes mellitus, pre-eclampsia, intrauterine fetal growth restriction, gestational diabetes, preterm birth, still birth, perinatal death and maternal morbidity and mortality; above all it adds nothing new to what is already known in the literature and moreover the same first author has already written 2 similar works on the topic-(Carmen-Ioana Marta in  diagnostic and biomedicines 2023)

Reviewer 4 Report

Comments and Suggestions for Authors

Prior to acceptance several points need to be addressed:

1. There is no information on the COVID-19
vaccination status of the patients (i.e. information of time points and number of vaccinations). As a COVID-19 vaccination uptake in Romania of more than 40% has been reported, vaccination of some patients is likely. Such information is particularly relevant as various adverse effects have been reported in hundreds of peer-reviewed publications and official information from government authorities.

1a. Meanwhile, it is quite safe to assume that the narrative of a successful COVID-19 vaccination campaign has collapsed. As a notable illustration, two papers are mentioned:

The excess mortality during 2020-2022 in 31 European countries as well as in Germany, respectively, has been statistically investigated in two studies by [Aarstadt and Kvitastein (2023) Is There a Link between the 2021 COVID-19 Vaccination Uptake in Europe and 2022 Excess All-Cause Mortality? Asian Pacific Journal of Health Sciences 10 (1), 25-31.] as well as [Kuhbandner and Reitzner (2023) Estimation of Excess Mortality in Germany During 2020-2022. Cureus 15(5):e39371.]  Both studies provide strong evidence that there is a covariation between the excess mortality and an increase in stillbirths (which is of particular interest for the present study)to the number of COVID-19 vaccinations, suggesting the possibility that mRNA COVID-19 vaccines were causatively involved in excess mortality.

1b. The authors mention the deaths of patients during the study, which should be specified, in particular, the vaccination status etc. This may require specific statistics as well.

2. The authors rely on and refer to COVID-19 viral infection identified through RT-PCR test, which is not a valid diagnostic tool (though officially implemented) as e.g. the inventor of the PCR method (Kary Mullis)had clearly stated. In line with the conclusion that a huge number of false negatives are expected, there have been numerous "asymptomatic infections" worldwide that represent nothing but essentially non-diseased persons. The latter is particularly relevant as the authors indicate such asymptomatic infections. These cases need to be separately investigated in the study.

3. The meaning of the follwing sentence (l. 312-314) is unclear: "Deceased women in our study had the highest NT-proBNP values among the studied patients, but those were not the highest values."

Comments on the Quality of English Language

There are no language issues.

Round 2

Reviewer 3 Report

Comments and Suggestions for Authors

With recent changes now the manuscript could be accepted, although it is not very original and despite the limitation of not having a more exhaustive correlation to multiple maternal and obstetric risk factor .

However, regarding infectious problems during pregnancy more generally, I suggest adding 2 bibliographical entries after the sixth-Introductione line 52:

A Vimercati, P Greco, G Loverro, Pl Lopalco, V Pansini, L Selvaggi. Maternal Complications After Caesarean Section In Hiv-Infected Women. Europ J Obstet Gynecol Repr Biol (90) 2000:73-76
Floridia M, Mastroiacovo P, Tamburrini E, Tibaldi C et Al,Italian Group On Surveillance On Antiretroviral Treatment In Pregnancy. “Birth Defects In A National Cohort Of Pregnant Women With HIV Infection In Italy, 2001-2011. BJOG. 2013;120(12):1466-75

I

Reviewer 4 Report

Comments and Suggestions for Authors

The resubmitted version essentially reflects a minor revision with little added information. The most important improvement is the notion that the patients were not vaccinated (and which offers important options for comparison with cohorts of vaccinated patients from other studies) However, most of the information concerning the health status, as demanded by the previous reviewer who recommended rejection has not yet been added.

Comments on the Quality of English Language

There are no lanuguage issues.

Round 3

Reviewer 4 Report

Comments and Suggestions for Authors

The points raised by the reviewers have been sufficiently addressed.